# Identification of an early subset of cerebellar nuclei neurons in mice

**Maryam Rahimi-Balaei[1,2], Shayan Amiri[1,3], Thomas Lamonerie[4], Sih-Rong Wu[5,6,7], Huda Y Zoghbi[6,7,8], G Giacomo Consalez[9,10], Daniel Goldowitz[2], Hassan Marzban[1]\***

[1]Department of Human Anatomy and Cell Science, The Children's Hospital Research Institute of Manitoba (CHRIM), Max Rady College of Medicine, Rady Faculty of Health Sciences, University of Manitoba, Winnipeg, Canada; [2]Department of Medical Genetics, Centre for Molecular Medicine and Therapeutics, BC Children's Hospital Research Institute, University of British Columbia, Vancouver, Canada; [3]Department of Pharmacology and Therapeutics, Division of Neurodegenerative Disorders, St Boniface Hospital Albrechtsen Research Centre, University of Manitoba, Winnipeg, Canada; [4]Université Côte d'Azur, CNRS, Inserm, iBV, Nice, France; [5]University of California, San Francisco (UCSF), San Francisco, United States; [6]Department of Neuroscience, Baylor College of Medicine, Houston, United States; [7]Jan and Dan Duncan Neurological Research Institute at Texas Children's Hospital, Houston, United States; [8]Howard Hughes Medical Institute, Baylor College of Medicine, Houston, United States; [9]Division of Neuroscience, IRCCS Ospedale San Raffaele, Milan, Italy; [10]Università Vita-Salute San Raffaele, Milan, Italy

**\*For correspondence:**
hassan.marzban@umanitoba.ca

**Competing interest:** The authors declare that no competing interests exist.

## eLife Assessment

The authors are interested in the developmental origin of the neurons of the cerebellar nuclei. In this study, they identify a population of neurons with a specific complement of markers that originate in a distinct location from where cerebellar nuclear precursor cells have been thought to originate that show distinct developmental properties. The discovery of a new germinal zone giving rise to a new population of neurons is an exciting finding, and it enriches our understanding of cerebellar development. The **important** claims, better explained in the current version, are well supported by **solid** evidence with the authors using a wide range of technical approaches, including transgenic mice that allow them to disentangle the influence of distinct developmental organizers

**Abstract** Cerebellar nuclei (CN) neurons serve as the primary output of the cerebellum and originate from the cerebellar primordium at early stages of cerebellar development. These neurons are diverse, integrating information from the cerebellar cortex and relaying it to various brain regions. Employing various methodologies, we have characterized a specific subset of CN neurons that do not originate from the rhombic lip or ventricular zone of the cerebellar primordium. Embryos were collected at early stages of development and processed for immunohistochemistry (IHC), western blotting, in situ hybridization (ISH), embryonic culture, DiI labeling, and flow cytometry analysis (FCM). Our findings indicate that a subset of CN neurons expressing α-synuclein (SNCA), OTX2, MEIS2, and p75NTR (NGFR) are located in the rostroventral region of the NTZ. While CN neurons derived from the rhombic lip are positioned in the caudodorsal area of the NTZ in the cerebellar primordium. Utilizing Otx2-GFP and *Atoh1*[-/-] mice, we have determined that these cells do not originate from the germinal zone of the cerebellar primordium. These results suggest the existence of a novel extrinsic germinal zone for the cerebellar primordium, possibly the mesencephalon, from which early CN neurons originate.

## Introduction

The cerebellum, a key player in both motor and non-motor functions, is a complex structure composed of the cerebellar nuclei (CN) and cerebellar cortex. The CN, the main cerebellar output to the rest of brain, are comprised of at least three distinct neuronal types: glutamatergic, GABAergic, and glycinergic neuron (*Koziol et al., 2014*; *Manto et al., 2012*; *Marzban et al., 2014*). In the mouse, the cerebellar primordium, an early structure in cerebellar development, emerges at approximately embryonic day (E)7–8 from the alar plate of the rhombomere-1 (*Kebschull et al., 2024*; *Goldowitz and Hamre, 1998*; *Sotelo, 2004*; *Wang and Zoghbi, 2001*; *Marzban et al., 2014*). The cerebellar primordium is rostrally limited by the isthmus, a boundary between the mesencephalon (midbrain)-rhombencephalon (hindbrain). Two key genes, orthodenticle homeobox 2 (*Otx2*) and gastrulation brain homeobox 2 (*Gbx2*), are among the earliest genes expressed in the neuroectoderm, dividing it into anterior and posterior domains with a border that marks the isthmus. *Otx2* is crucial for the development of the forebrain and midbrain, while *Gbx2* is essential for the anterior hindbrain (*Millet et al., 1999*). The isthmus, where Otx2 and Gbx2 expression meets, is a multi-signaling center that regulates other genes essential for the early development of both the midbrain and cerebellar primordium (*Joyner et al., 2000*; *Wurst and Bally-Cuif, 2001*).

The cerebellar primordium contains two distinct germinal zones, which are responsible for cerebellar neurogenesis and gliogenesis: the ventrally located ventricular zone (VZ) and dorsally located rhombic lip (RL) (*Fink et al., 2006*). Early studies on the development of CN indicated that both glutamatergic and GABAergic CN neurons originate from the cerebellar VZ (*Voogd and Glickstein, 1998*). Further studies highlighted the involvement of the RL as an origin for glutamatergic CN neurons (between E9 and E12) (*Fink et al., 2006*; *Wang and Zoghbi, 2001*; *Green and Wingate, 2014*; *Machold and Fishell, 2005*; *Kebschull et al., 2024*). LIM homeobox transcription factor 1 alpha (LMX1A) regulates the development of the glutamatergic CN neurons originating from the RL. It is also expressed in the nuclear transitory zone (NTZ) and is considered to be a marker for the majority of rhombic lip-derived CN neurons in addition to a subset of LMX1A-positive cells that do not originate from the RL migratory stream (*Chizhikov et al., 2006*; *Chizhikov et al., 2010*).

Extensive developmental studies have shown that all cerebellar cells are produced sequentially from the VZ and the upper RL (*Marzban et al., 2014*). In this report, we provide evidence that a subset of CN neurons expressing OTX2, α-synuclein (SNCA), MEIS2, and p75 neurotrophin receptor (p75NTR)/nerve growth factor receptor (NGFR) originate from the rostral end of the cerebellar primordium and are positioned in the rostroventral region of the NTZ during early cerebellar development. Our data suggest that the mesencephalon represents a third germinal zone and is the origin of previously unrecognized neurons that contribute to CN formation and development.

## Results

In mouse embryos, the earliest neuronal populations, which are located at the rostral end of the cerebellar primordium, are immunopositive for neurofilament-associated antigens (NAA; detected by 3A10 antibody), a marker typically expressed in neuronal somata and axons (*Marzban et al., 2008*; *Marzban et al., 2019*; *Figure 1*). To further explore these cells in the rostral end of the cerebellar primordium at E9, we used an antibody to detect SNCA, which is expressed in CN neurons during embryogenesis (*Zhong et al., 2010*). We identified SNCA⁺ neurons at the rostral end of the NTZ in the cerebellar primordium (*Figure 1E–F* and *Figure 5—figure supplement 1*). At E12, we aimed to determine the location of the SNCA⁺ cells in the cerebellar primordium and compare their position with that of the rhombic lip-derived LMX1A⁺ CN neurons (E10–12) in the NTZ. To perform this, double immunofluorescence labeling of SNCA and LMX1A (*Figure 2A–D*, medial section and *Figure 2E–H*, lateral section) showed that an LMX1A⁺ population of CN neurons, originating from the RL, either flanks SNCA⁺ neurons in the medial section (*Figure 2A–D*) or is located dorsally in the lateral section (*Figure 2E–H*) in the NTZ. This indicates that SCNA⁺ cells are a distinct cell population different from rhombic lip-derived LMX1A⁺ cells. To further explore the characteristics of SNCA⁺ cells, we wanted to determine whether these cells originate from the mesencephalon. OTX2, which is highly expressed in the mesencephalon, has its caudal limit at the boundary of the rhombencephalon (i.e. the isthmus) (*Kurokawa et al., 2004*), thus makes it an appropriate marker for identifying mesencephalon-derived cells. Interestingly, results from double immunofluorescence experiments in sagittal sections of E12

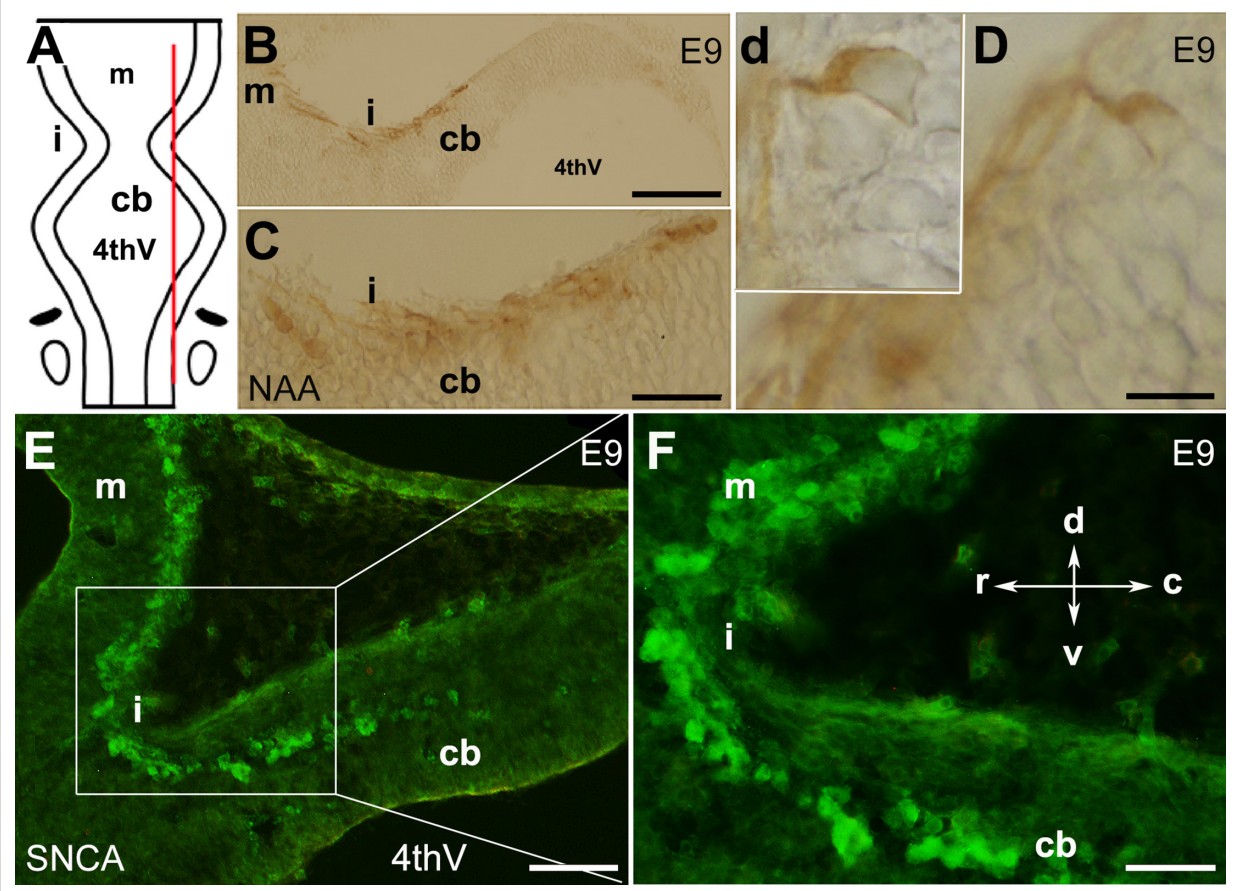

**Figure 1.** The presence of neurons in the cerebellar primordium at embryonic day (E)9 and spatial expression of SNCA. The cerebellar primordium immunostained with neurofilament-associated antigens (NAA) and SNCA shows that a distinct subset of neurons located at the rostral end of the developing cerebellum. (**A**) Dorsal view of the schematic illustration of the cerebellar primordium, mesencephalon, isthmus, and 4th ventricle. The red line indicates the sagittal plane about which the sections shown in (**B–D**) were taken. (**B–D**) Sagittal section through the cerebellar primordium at early E9 immunoperoxidase-labeled with NAA 3A10 shows the presence of neurons in the cerebellar primordium that cross the isthmus (i) and continue to the mesencephalon. (**C**) A higher magnification of (**B**). (**D**) Differentiated neurons at E9.5 are visible; a higher magnification is shown in the inset, **d**. (**E**) Sagittal section through the cerebellar primordium at late E9. Immunofluorescence labeling of SNCA shows SNCA$^+$ (green) expressing CN neurons in the mesencephalon, at the isthmus (i) and in the rostral part of the cerebellar primordium (cb). (**F**) A higher magnification of (**E**). Abbreviations: 4thv, 4th ventricle; cb, cerebellum; c, caudal; d, dorsal; i, isthmus; m, mesencephalon; r, rostral; v, ventral. Scale bar = 100 μm in (**B**); 50 μm in (**C** and **E**); 20 μm in (**D**).

cerebella showed that OTX2 was highly expressed in the mesencephalon (*Figure 3A and B*), and was co-expressed with SNCA$^+$ cells in the rostral region of the cerebellar primordium (*Figure 3A–E*). This result suggests that SNCA$^+$ cells originate from the mesencephalon. Next, we used quantitative flow cytometry analysis (FCM) to determine whether OTX2$^+$ cells co-express SNCA and/or LMX1A (*Figure 3F and G*). An increase in the number of OTX2- and SNCA-positive cells (but not LMX1A) was observed at E14 in comparison to E12 (*p<0.05). Interestingly, the majority of neural cells were not SNCA$^+$/LMX1A$^+$, consistent with our IHC findings, or SNCA$^+$/OTX2$^+$, with the exception of a few cells that did show co-labeling. Of note, more SNCA$^+$/OTX2$^+$ than SNCA$^+$/LMX1A$^+$ cells were detected at E12 compared to E14, indicating the co-expression of these markers at different time point during cerebellar development. An increase in the OTX2 expression at the rostral end of the cerebellar primordium was also confirmed by the IHC at E12–15 (*Figure 4A–F*). The total number of OTX2$^+$ cells in the cerebellar primordium sections was counted from E12 to E15 (*Figure 4G*) and comparison between these embryonic stages indicated a slight increase in the number of OTX2$^+$ cells between E12 and E15. Significant differences were observed between E12 and E14 (**p<0.01), E12 and E15 (***p<0.001), as well as E13 and E15 (#p<0.05).

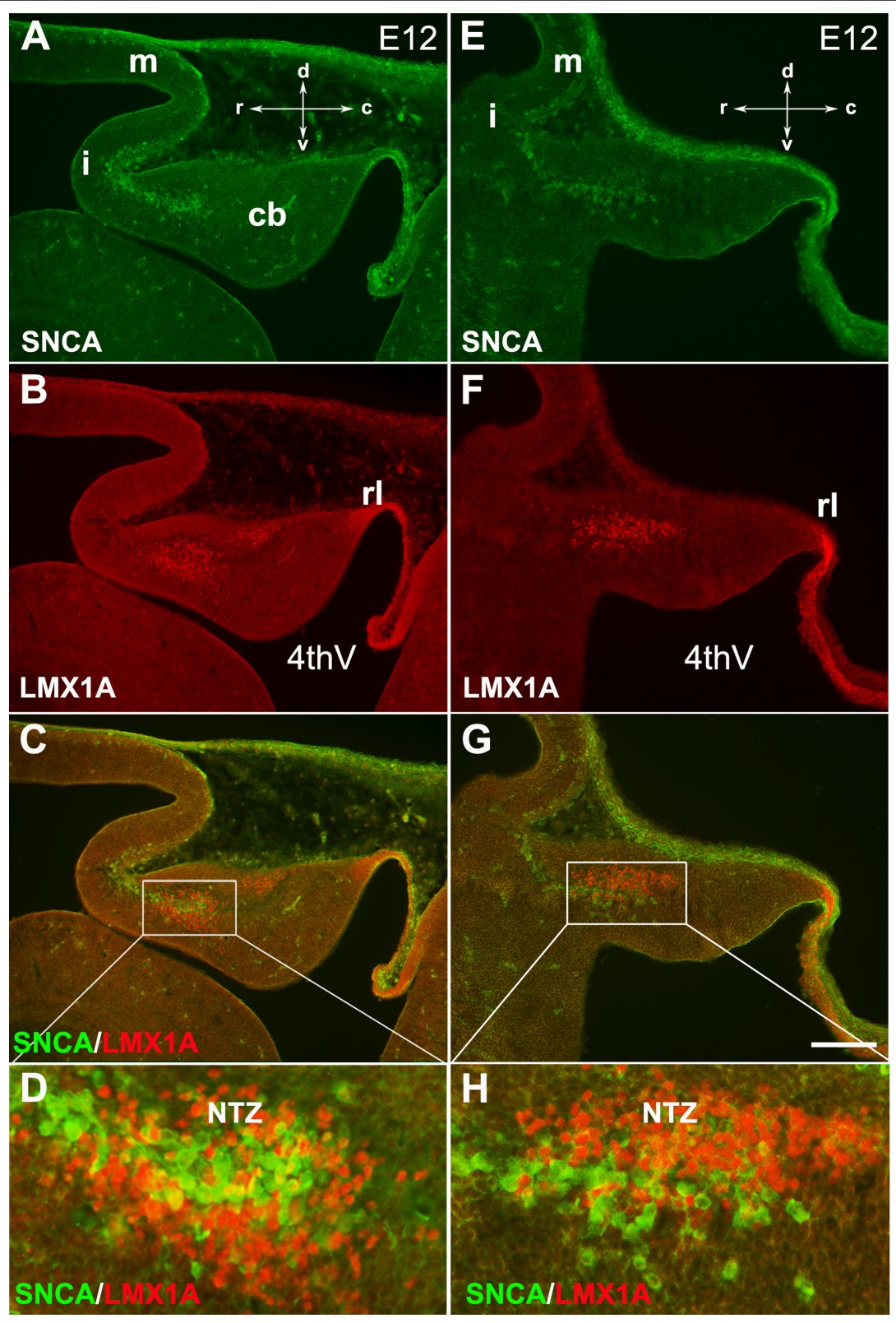

**Figure 2.** SNCA expression relative to Lmx1a in the early developing mouse cerebellum. The sagittal section through the cerebellar primordium at embryonic day (E)12.5 reveals double immunofluorescence labeling with SNCA and LMX1A antibodies in both medial (**A–D**) and lateral (**E–H**) sections. (**A–G**) SNCA (green, **A and E**) and LMX1A (red, **B and F**) immunopositive cells are located at the NTZ. SNCA+ cells continue to the mesencephalon, and LMX1A+ cells toward the rhombic lip. Merged images show that the SNCA+ cells form a population of cerebellar nuclei (CN) neurons distinct from

*Figure 2 continued on next page*

Figure 2 continued

the rhombic lip-derived cells (LMX1A+) in the NTZ (**C and G**). (**D** and **H**) show a higher magnification of (**C** and **G**), respectively. Abbreviations: 4thV, 4th ventricle; cb, cerebellum; c, caudal; d, dorsal; i, isthmus; m, mesencephalon; rl, rhombic lip; NTZ, nuclear transitory zone; r, rostral; v, ventral. Scale bar = 100 µm.

To further confirm the presence of OTX2+ cells in the rostral end of the cerebellar primordium, we used *Otx2*-GFP transgenic mice and performed double immunofluorescence labeling with OTX2 and GFP on E13 sections (***Figure 5***). Results showed that an extension of OTX2-GFP-positive cells, highly prevalent in the mesencephalon, crossed the isthmus and terminated in the rostral end of the cerebellar primordium in the NTZ (***Figure 5B, C, and D***). To further validate our observations, we employed RNAscope fluorescence in situ hybridization (FISH) at E12 to detect the presence of *Otx2* mRNA in the cerebellar primordium (***Figure 5E–J***). *Otx2* mRNA was highly expressed in the mesencephalon and caudally extended to the rostral cerebellar primordium in the NTZ (***Figure 5E–J*** and ***Figure 5—figure supplement 1***).

To evaluate whether OTX2+ cells are present in the absence of *Snca* gene expression, we used *Pap⁻/⁻*; *Snca⁻/⁻* mice. OTX2 immunoperoxidase staining showed that even in the absence of *Snca* expression, OTX2+ cells still terminated in the rostral cerebellar primordium (***Figure 3—figure supplement 1***). This indicates that the expression of Otx2 in these cells is independent of SNCA. We previously reported that NGFR (p75NTR; *Ngfr*), a neuronal proliferation and differentiation marker, was expressed in the rostral end of the cerebellar primordium in *Pap⁻/⁻*; *Snca⁻/⁻* mice (***Jiang et al., 2009***; ***Bernabeu and Longo, 2010***; ***Dechant and Barde, 2002***; ***Rahimi-Balaei et al., 2019***). To determine whether SNCA+ neurons are p75NTR immunopositive, double immunolabeling with SNCA and p75NTR revealed that SNCA+ cells in the NTZ exhibited cell membrane expression of p75NTR (***Figure 6A–D***). This data was further confirmed by western blot analysis of SNCA and p75NTR protein expression in embryonic cerebellar lysates (E11, E13, and E15; ***Figure 6D***). In addition, to conclusively determine that p75NTR is expressed in the membranes of SNCA+ cells, we utilized primary dissociated cultures of cerebellum at E10, days in vitro (DIV) 4 (***Figure 6E–H***). IHC demonstrated that p75NTR was indeed expressed in the membranes of SNCA+ cells (with punctate appearance, ***Figure 6E–H***; ***Dechant and Barde, 2002***).

Atonal BHLH transcription factor 1 (ATOH1) plays a crucial role in regulating the development of glutamatergic CN neurons derived from the RL at E12, prior to the formation of other rhombic lip-derived cells (granule cells and unipolar brush cells) (***Ben-Arie et al., 1997***; ***Wang et al., 2005***). To determine whether OTX2 and SNCA immunopositive cells exist in the absence of rhombic lip-derived CN neurons, we used Atoh1⁻/⁻ mice (***Figure 7A–C***). Immunostaining revealed the presence of SNCA+ and OTX2+ cells at the rostral end of the cerebellar primordium at E12, which reveals this cell population has no RL origin. To validate our findings, we evaluated MEIS2, a transcription factor typically expressed in the NTZ and glutamatergic CN neurons (***Willett et al., 2019***), in Atoh1+/+ and Atoh1⁻/⁻ cerebellar sections at E12. Notably, our results showed the presence of two distinct sets of MEIS2+ cells in the NTZ of Atoh1+/+ embryos: (1) rhombic lip-derived CN neurons located in the caudodorsal region, which do not express SNCA (MEIS2+/SNCA⁻), and (2) a subset of MEIS2+/SNCA+ CN neurons situated in the rostroventral region of the NTZ (***Figure 7D–I***). In Atoh1⁻/⁻ sections, MEIS2+/SNCA+ cells were still present in the rostroventral region of the NTZ, whereas the subpopulation of MEIS2+/SNCA⁻ rhombic lip-derived CN neurons was absent in the caudodorsal region of the NTZ (***Figure 7J–O***).

To explore if the rostroventral subset of CN neurons derived from an external cerebellar germinal zone, possibly the mesencephalon, we utilized FAST DiI as a neuronal tracer to mark cells within a potential source region. FAST DiI was applied to the dorsum of the caudal mesencephalon at E9, where it was maintained for 4 days (***Figure 8A, a***). At DIV 4, cells stained with DiI in the dorsum of the mesencephalon were present in both rostral (***Figure 8B***) and caudal (***Figure 8C***) directions. Sections from the cerebellar primordium revealed the presence of DiI-labeled cells within the cultured embryo (***Figure 8—figure supplement 1***). To investigate the earliest DiI-positive cell population in the mesencephalon and avoid unwanted cell staining due to long DiI exposure, we focused on early cerebellar development at E9; it has been reported that one of the premier neuronal populations in the CNS is present at this time in the mesencephalon and projects caudally (***Stainier and Gilbert, 1990***; ***Easter et al., 1993***). To determine if an early generation of mesencephalic DiI-labeled cells is present within the cerebellar primordium, we limited cellular DiI exposure to only 24 hr (***Figure 8D, d***), after which it was removed (***Figure 8—figure supplement 2***). Intriguingly, almost all DiI+ cells migrated caudally

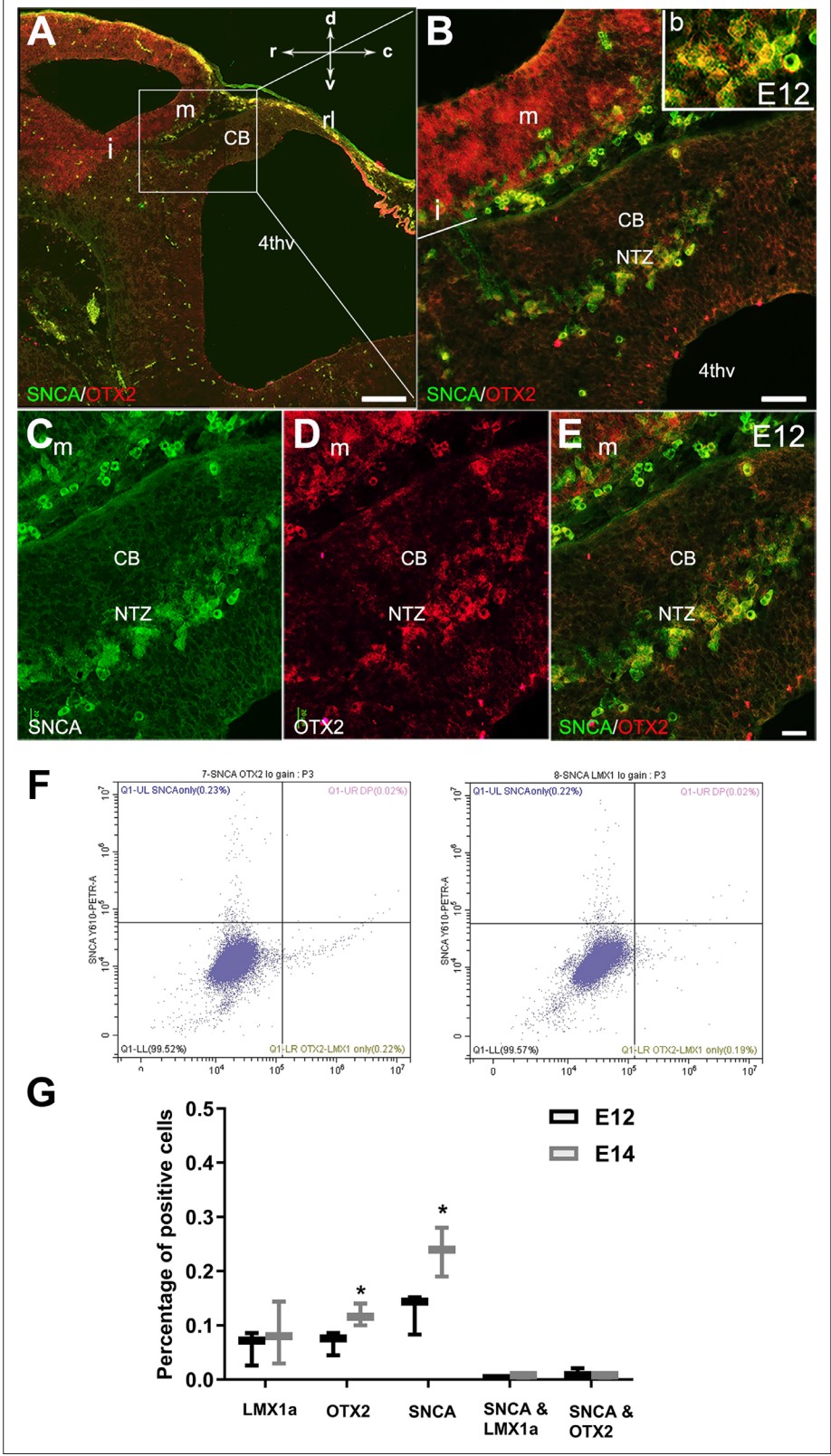

**Figure 3.** Expression of SNCA and OTX2 in the developing cerebellar primordium. Double immunofluorescence labeling with SNCA and OTX2 antibodies in sagittal sections of the cerebellar primordium at embryonic day (E)12.5. (**A–E**) Double immunolabeling of SNCA (green) and OTX2 (red) of a sagittal section of the cerebellar primordium at E12.5. (**A**) High OTX2 immunoreactivity was detected in the mesencephalon. SNCA⁺ cells in the

*Figure 3 continued on next page*

*Figure 3 continued*

mesencephalon accompany OTX2$^+$ cells across the isthmus (i) and in the NTZ. (**B**) Higher magnification of (**A**). (**C–E**) Immunolabeling with SNCA (green, **C**) and OTX2 (red, **D**) and merged (**E**) shows co-expression in the NTZ. (**F–G**) Cerebellar cells at E12 and E14 were dissociated and immunolabeled for SNCA+OTX2, SNCA+LMX1A, or each antibody individually (**F**). Quantitative analysis was performed by flow cytometry to detect SNCA$^+$, OTX2$^+$, LMX1A$^+$, SNCA$^+$/ LMX1A$^+$, and SNCA$^+$/OTX2$^+$ cells (**G**). No significant differences were observed for the number of LMX1A$^+$ cells between E12 and E14. An increase in SNCA- and OTX2-positive cells was shown at E14 in comparison to E12 (*$p<0.05$). The number of cells positive for SNCA/LMX1A or SNCA/OTX2 was very low at E12 and E14. Abbreviations: 4thv, 4th ventricle; cb, cerebellum; c, caudal; d, dorsal; i, isthmus; m, mesencephalon; rl, rhombic lip; NTZ, nuclear transitory zone; r, rostral; v, ventral. Scale bar = 200 μm in (**A**); 50 μm in (**B**); 20 μm in (**C, D** and **E**).

The online version of this article includes the following source data and figure supplement(s) for figure 3:

**Source data 1.** The content of this file is original FACS data for *Figure 3F*.

**Source data 2.** The content of this file is used for *Figure 3G*.

**Figure supplement 1.** OTX2 expression persists in Snca null mice.

and presented in the cerebellar primordium (*Figure 8E, e and Figure 8—figure supplement 2*). DiI staining was clear and strong in cells as shown in *Figure 8*, *Figure 8—figure supplement 1*; *Figure 2* (arrowhead); it should be noted, however, that some DiI staining was evident in other cells in the rostral end of the cerebellar primordium, although the signal appeared relatively weak (*Figure 8— figure supplement 2*).

## Discussion

Previously, it was commonly held that all CN neurons had a singular origin in the VZ (*Pierce, 1975*; *Miale and Sidman, 1961*). However, recent discoveries have challenged this notion, revealing a dual origin for CN neurons. Some originate from the VZ, while others arise from the RL (*Wang and Zoghbi, 2001*; *Wang et al., 2005*; *Ben-Arie et al., 1997*). Genetic fate mapping further suggested that gluta-matergic CN projection neurons may arise from the RL (*Ben-Arie et al., 1997*; *Machold and Fishell, 2005*; *Fink et al., 2006*). Transcription factor expression patterns indicate that CN neurons migrate from the RL to the NTZ through a subpial stream pathway, while sequentially expressing the genes *Lmx1a*, *Pax6*, *Tbr2,* and *Tbr1* (*Fink et al., 2006*).

In this study, we characterized a subset of CN neurons that do not originate from the cerebellar primordium germinal zones (RL and VZ) during early cerebellar development. We demonstrated a newly identified subset of CN neurons, which are SNCA$^+$/OTX2$^+$/MEIS2$^+$/p75NTR$^+$/LMX1A$^-$, in the rostroventral region of the NTZ of the cerebellar primordium. This suggests the existence of a new germinal zone during cerebellar neurogenesis.

The exact origin of the SNCA$^+$/OTX2$^+$/MEIS2$^+$/p75NTR$^+$/LMX1A$^-$ CN neurons is currently unclear; however, they do not originate from the RL or VZ. RL-derived TBR1$^+$/LMX1A$^+$ CN neurons are born at E9, but do not reach the NTZ until ~E11 (*Kebschull et al., 2024*; *Machold and Fishell, 2005*; *Wang et al., 2005*; *Manto et al., 2012*; *Fink et al., 2006*; *Rahimi-Balaei et al., 2018*). Our results revealed that SNCA$^+$ cells are a group of differentiating neurons (NAA 3A10$^+$) present in the NTZ at E9, before the arrival of any neurons that originate from the RL. The majority of SNCA$^+$ CN neurons are not LMX1A$^+$; however, a small proportion of cells seems to exhibit co-expression. This suggests that in the early stages of CN neurogenesis, the pattern of protein expression in SNCA$^+$ neurons may be changing, possibly with respective down- and upregulation of *Snca* and *Lmx1a*. Conversely, this may not occur at all, and 'co-labeling' could have been observed due to overlapping cells. A similar display of SNCA expression has recently been demonstrated in oligodendrocyte development and maturation (*Djelloul et al., 2015*). Our findings indicated that CN neuron somata are highly positive for SNCA until E14, after which SNCA expression is refined to Purkinje cells at P0 (*Zhong et al., 2010*) and axonal projections, suggesting that SNCA$^+$ cells at the rostral end of the cerebellar primordium are associated with CN establishment in the NTZ. Our results from studies using Pap$^{-/-}$; *Snca$^{-/-}$* mice showed that lack of SNCA during cerebellar development did not affect the formation of rostroven-tral subset of CN neurons (*Rahimi-Balaei et al., 2019*), which still expressed OTX2 and terminated in the NTZ. While our comprehensive fate mapping study of SNCA-expressing CN neurons is still in

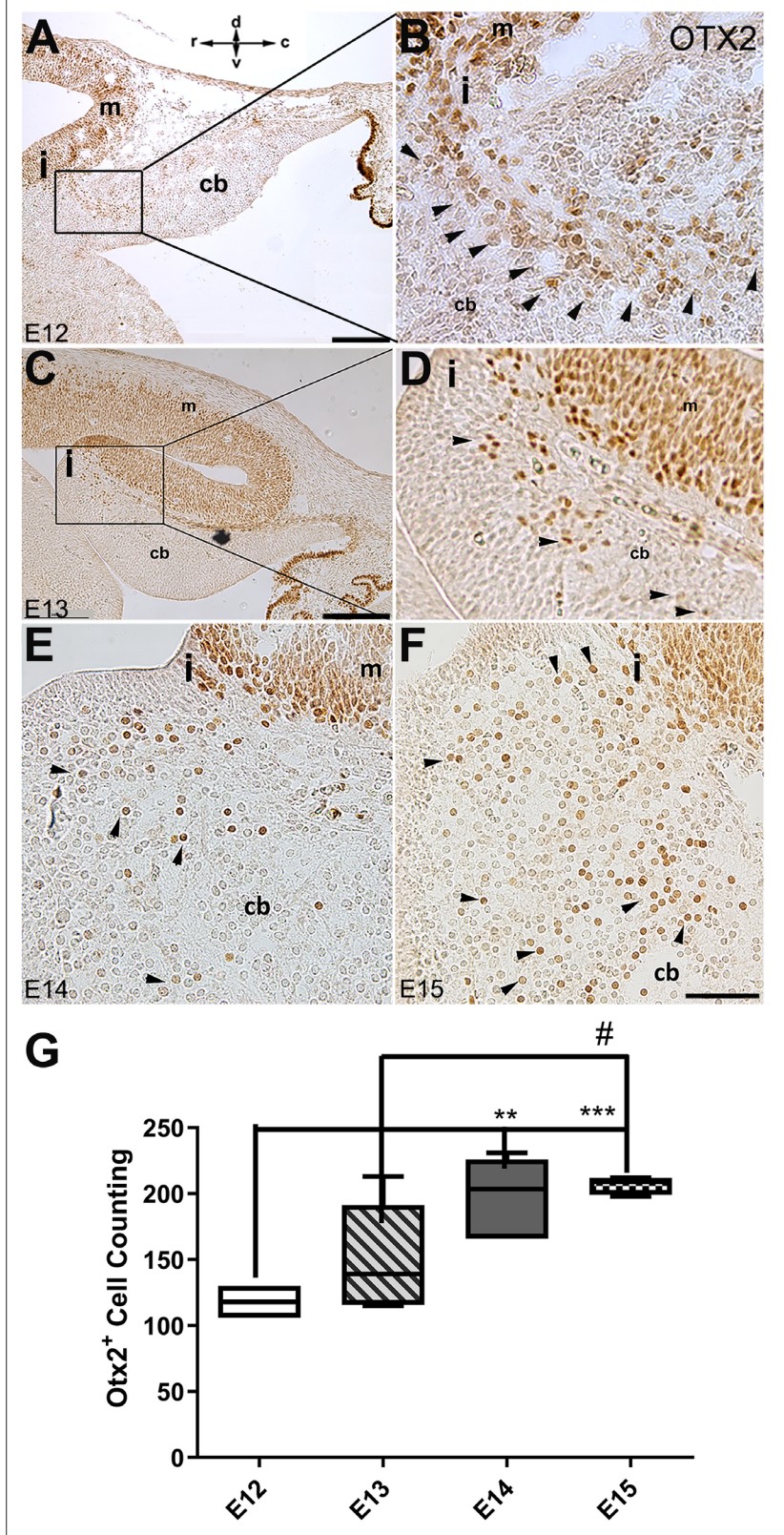

**Figure 4.** The spatial and temporal expression of OTX2 in the developing cerebellum of mice. Sagittal sections through the cerebellar primordia at E12, E13, E14, and E15 were analyzed for peroxidase immunoreactivity of OTX2. (**A–F**) Sagittal section through cerebellar primordium at E12 (**A**; higher magnification shown in **B**), E13 (**C**; higher magnification shown in **D**), E14 (**E**), and E15 (**F**). High OTX2 immunoreactivity at the mesencephalon

*Figure 4 continued on next page*

*Figure 4 continued*

is evident, and a few OTX2⁺ cells cross the isthmus and position at the rostral part of the cerebellar primordium at the NTZ. Examples of Otx2 expressing cells are indicated by arrowheads in **B, D, E, and F**. In (**E and F**), the isthmus and adjacent cerebellar area are shown in the same orientation as (A–D) (I; isthmus). The orientation label (r-c and d-v; rostral caudal and dorsal ventral) in (**A**) applies to all panels. (**G**) Comparison of the number of OTX2-positive cells in the cerebellar primordium from E12 to E15. Results indicate a slight increase in the number of OTX2-positive cells over time. Significant differences were observed between E12 and E14 (**\*\*p<0.01**), E12, and E15 (**\*\*\*p<0.001**), as well as E13 and E15 (**#p<0.05**). Data were analyzed by one-way ANOVA followed by a Tukey's multiple comparison test. Abbreviations: 4thv, 4th ventricle; cb, cerebellum; c, caudal; d, dorsal; i, isthmus; m, mesencephalon; rl, rhombic lip; r, rostral; v, ventral. Scale bar = 200 μm in (**A**) and (**C**); 50 μm in (**F**) (applies to **B, D, E**, and **F**).

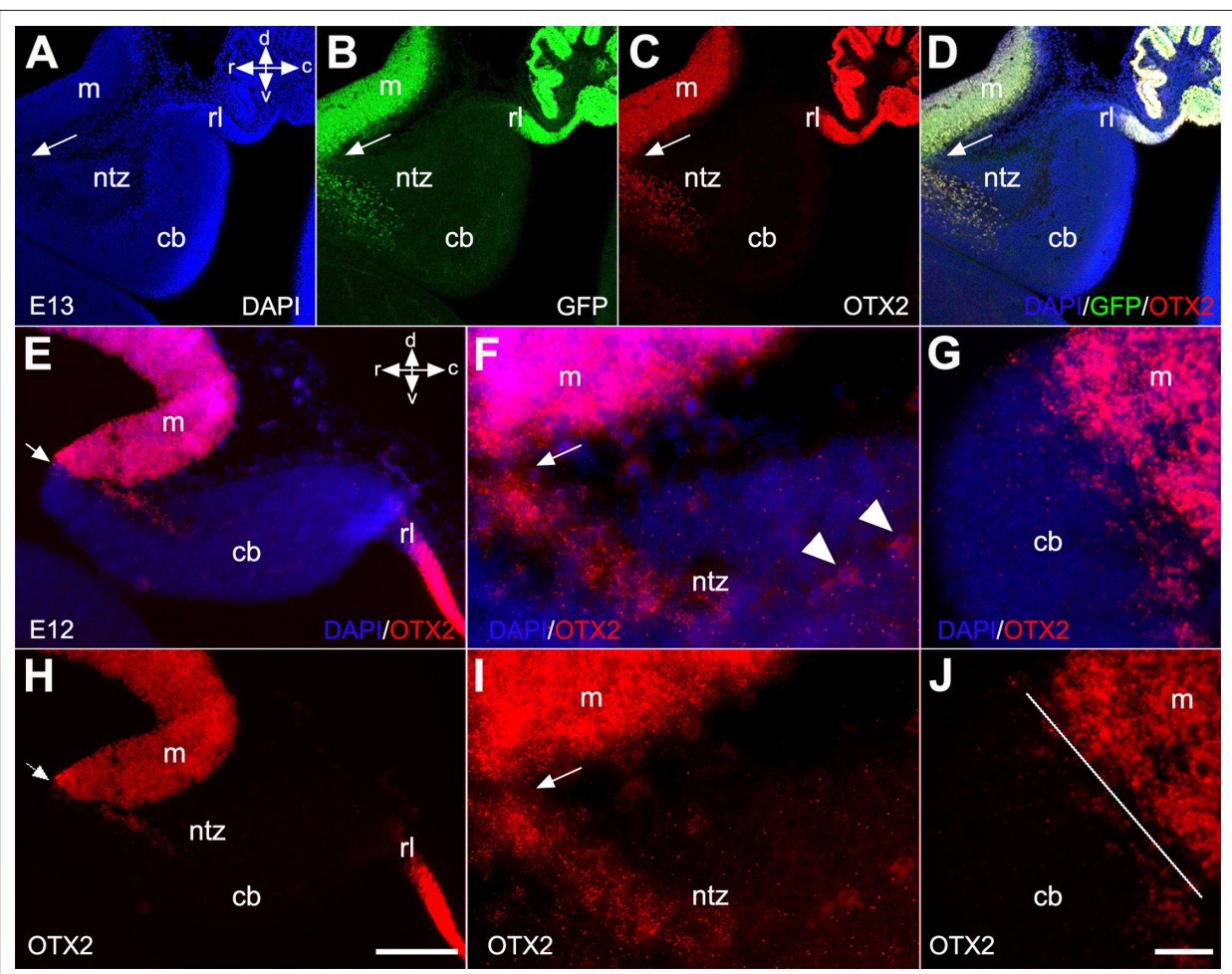

**Figure 5.** Otx2 expression in developing cerebella of Otx2-GFP transgenic and control mice. Sagittal sections through the cerebellar primordium of Otx2-GFP transgenic mice at embryonic day (E)13 are shown in (**A–D**), and RNAscope fluorescence in situ hybridization (FISH) of Otx2 at E12 in wild-type mice in (**E–J**). (**A–D**) DAPI labels the outline of the cerebellar primordium and mesencephalon (**A**, blue). GFP expression, which is enhanced by immunofluorescence labeling with anti-GFP (**B**, green), and immunofluorescence labeling for OTX2 (**C**, red), reveal co-labeled cells (**D**, merged) in the mesencephalon and NTZ at the rostral end of the cerebellar primordium. Arrows indicate the isthmus. (**E–G**) Merged channels of the in situ hybridization of *Otx2* mRNA probe counterstained with DAPI at low (**E**) and high (**F and G**) magnification captured by confocal microscopy. (**H–J**) The Otx2 mRNA signal was strong in the mesencephalon and extended as a tail through the isthmus to the rostral cerebellar primordium in the NTZ. The isthmus is indicated by arrows in (**E, F, H, I**) and a line in (**J**). Abbreviations: cb, cerebellum; c, caudal; d, dorsal; m, mesencephalon; rl, rhombic lip; ntz, nuclear transitory zone; r, rostral; v, ventral. Scale bar = 200 μm in (**D**) (applies to **A–D**); 100 μm in (**H**) (applies to **E** and **H**); 20 μm in (**J**) (applies to **F–J**).

The online version of this article includes the following figure supplement(s) for figure 5:

**Figure supplement 1.** In situ hybridization demonstrates expression pattern of Otx2, Snca, C-Ret, and Tlx3.

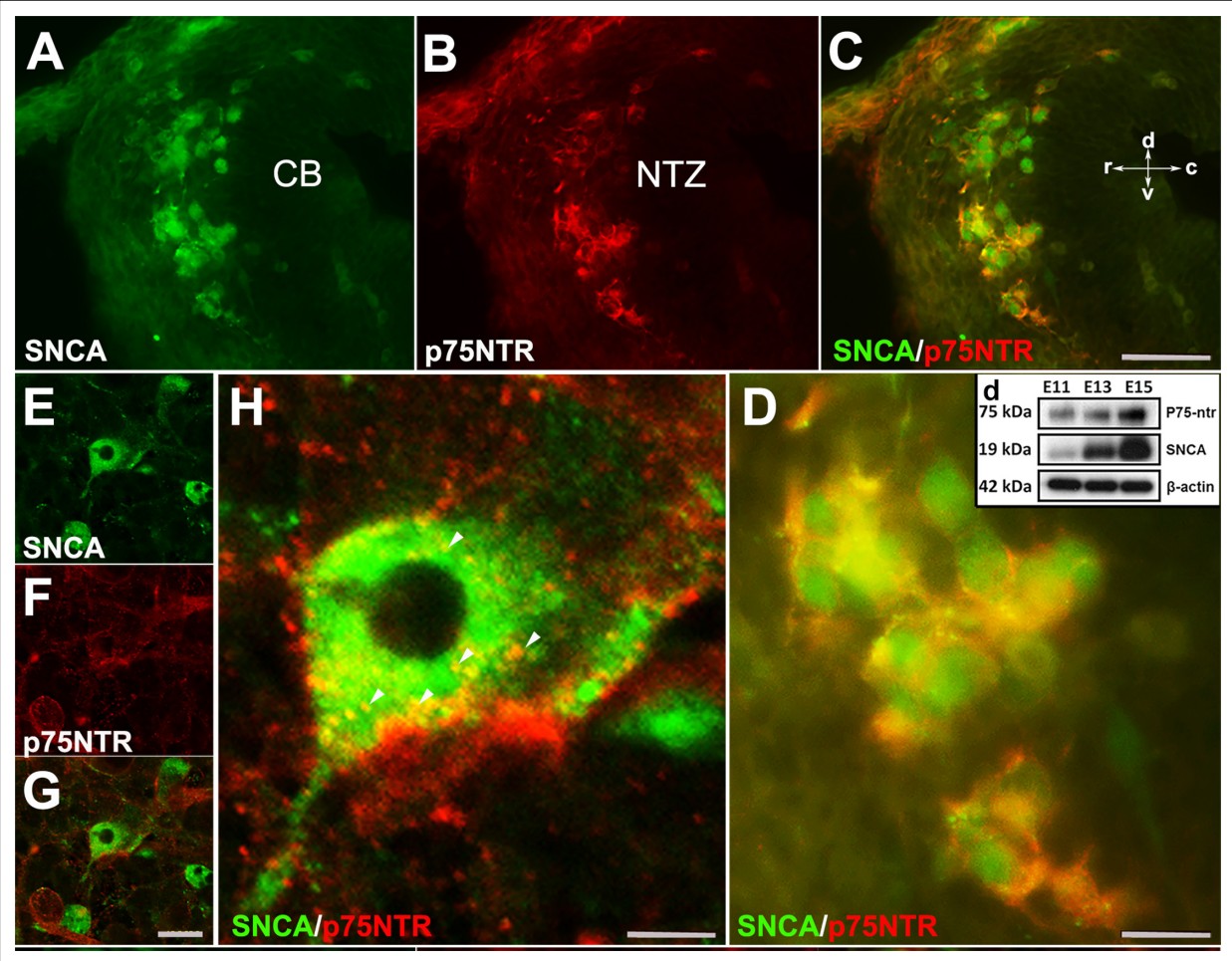

**Figure 6.** SNCA immunopositive cells express p75NTR (nerve growth factor receptor [NGFR]) in developing cerebellum in vivo and in vitro. (**A–D**) Double immunofluorescence labeling with SNCA **A**, green and p75NTR (**B**, red) antibodies reveals co-labeled cells (**C**, merged) in the NTZ; a higher magnification of the NTZ is shown in (**D**). (**d**) Western blot analysis of SNCA and p75NTR expression during cerebellar development. Immunoblots of total cerebellar lysates from embryos at E11, E13, and E15 indicate an increase in expression of SNCA and p75NTR from E11 to E15. Equal protein loading was confirmed by β-actin expression. (**E–H**) Primary dissociated cultures of cerebellum obtained from an E10 mouse embryo (days in vitro [DIV] 4), double immunofluorescence stained for SNCA (**E**: green) and p75NTR (**F**: red) (merged image shown in **G**). (**H**) is a higher magnification of (**G**); punctuate cellular p75NTR immunoreactivity is marked by arrow heads. Abbreviations: cb, cerebellum; c, caudal; d, dorsal; NTZ, nuclear transitory zone; r, rostral; v, ventral. Scale bar = 50 µm in (**A–C**) and (**D–F**); 20 µm in (**H**); 10 µm in (**G**).

The online version of this article includes the following source data for figure 6:

**Source data 1.** This file has the original scan of western blot bands used for *Figure 6D*.

**Source data 2.** The content of this file is showing replicates of western blot finding, supporting *Figure 6D*.

progress, previous research has documented SNCA expression in the medial CN neurons of adult mice. (*Fujita et al., 2020*). Excitatory CN neurons were previously classified into two main types. However, recent research has further subdivided these neurons into spatially distinct subpopulations within each CN (*Kebschull et al., 2024*; *Krishnamurthy et al., 2024*). Specifically, excitatory neurons in the medial CN have been divided into four subpopulations based on the expression of three marker proteins, including SNCA (*Fujita et al., 2020*). Notably, SNCA[+] neurons are predominantly located ventrally in the caudal medial CN. Further studies using mice with mutations in En1 and En2, which are crucial for regulating the function of the isthmic organizer, have demonstrated a loss of a subpopulation of excitatory neurons in the medial CN (*Krishnamurthy et al., 2024*). This finding suggests that the early SNCA[+] subpopulation may consist of excitatory neurons that did not originate from the RL.

Furthermore, our observations reveal the expression of MEIS2 in the NTZ (*Figure 7*) and as it is shown previously (*Morales and Hatten, 2006*). The existance of discrete subpopulations of precursors

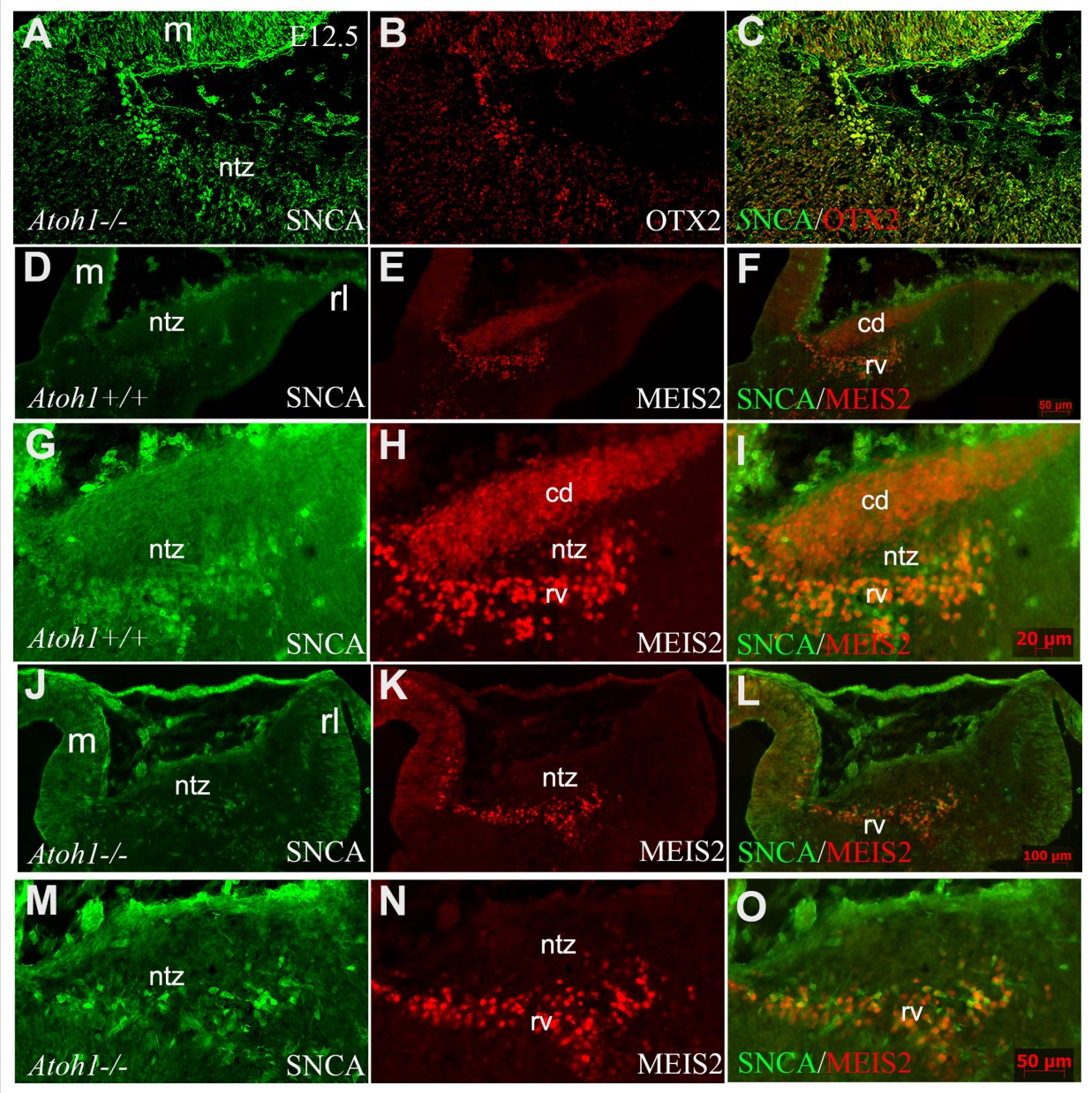

**Figure 7.** SNCA, OTX2, and MEIS2 expression in the developing cerebella of Atoh1 null and control mice. Sagittal sections through the cerebellar primordia of Atoh1$^{+/+}$ and Atoh1$^{-/-}$ embryos at embryonic day (E)12.5; double immunofluorescence labeling with SNCA+OTX2 and SNCA+MEIS2 antibodies is demonstrated. (**A–C**) SNCA (green, **A**) and OTX2 (red, **B**) immunopositive cells are present in the mesencephalon and NTZ of Atoh1 KO mouse (merged channels, **C**).(**D–F**) In E12.5 Atoh1$^{+/+}$ sagittal sections, SNCA$^{+}$ (green, **D**) cells are observed in the mesencephalon and NTZ. MEIS2$^{+}$ (red, **E**) cells are present in the mesencephalon and NTZ in addition to another population of MEIS2$^{+}$ cells in the dorsal region of the NTZ which extends to the rhombic lip. Merged image (**F**) confirms the presence of two distinct sets of MEIS2$^{+}$ cells in the NTZ: rhombic lip-derived CN neurons located in the caudodorsal (cd) region, which do not express SNCA (MEIS2$^{+}$/SNCA$^{-}$), and a subset of MEIS2$^{+}$/SNCA$^{+}$ CN neurons situated in the rostroventral (rv) region of the NTZ. (**G–I**) Higher magnification of (**D–F**). (**J–L**) In E12.5 Atoh1$^{-/-}$ sagittal sections, the expression of SNCA (green, **J**) shows no change as compared to Atoh1$^{+/+}$, whereas Meis2 abundance (red, **K**) exhibited a different pattern. Thus, MEIS2$^{+}$/SNCA$^{+}$ cells were still present in the rostroventral region of the NTZ. However, the subpopulation of MEIS2$^{+}$/SNCA$^{-}$ rhombic lip-derived CN neurons were absent in the caudodorsal region of the NTZ. (**M–O**) Higher magnification of (**J–I**). Abbreviations: 4thV, 4th ventricle; cb, cerebellum; cd, caudodorsal; i, isthmus; m, mesencephalon; rl, rhombic lip; NTZ, nuclear transitory zone; rv, rostroventral. Scale bar = 100 μm.

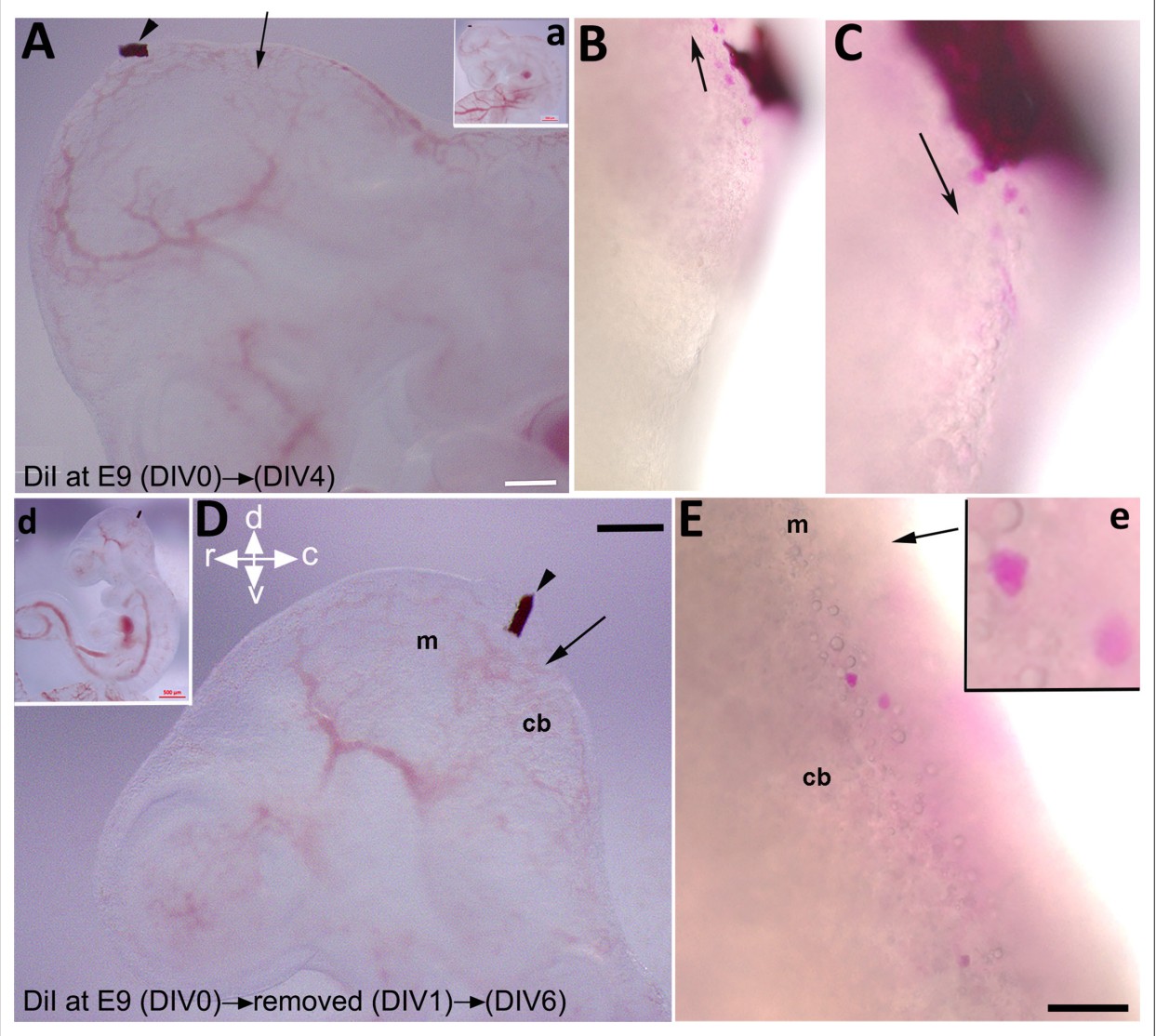

**Figure 8.** Labeling of mesencephalic cells during early development in mice using embryonic culture. FAST DiI was applied to an embryo at embryonic day (E)9 and kept for 4 days (days in vitro [DIV] 4, **A–C**) or removed after 24 hr (DIV 1, **D–E**). (**A, a**) FAST DiI was inserted in the mesencephalon at E9 (DIV 0), arrowhead shows insertion location of DiI crystal in the mesencephalon. The arrow indicates the isthmus. Inset 'a' shows the whole embryo and indicates the site of DiI insertion. (**B–C**) 4 days post DiI insertion, DiI-positive cells directed both rostrally to the mesencephalon (**B**) and caudally to the rostral cerebellar primordium (**C**). (**D, d**) FAST DiI was inserted in the mesencephalon at E9 (DIV 0) (indicated by arrowhead) and removed at DIV 1, arrow shows the isthmus. Inset 'd' shows the whole embryo and indicates the site of DiI insertion. (**E, e**) DiI-positive cells present in cerebellar primordium at DIV 6 after removal of FAST DiI at DIV 1. Abbreviations: cb, cerebellum; c, caudal; d, dorsal; m, mesencephalon; r, rostral; v, ventral. Scale bar = 500 µm in (**a, d**); 200 µm in (**A, D**); 100 µm in (**B–C, E**).

The online version of this article includes the following figure supplement(s) for figure 8:

**Figure supplement 1.** FAST DiI was applied to an embryo at embryonic day (E9) and kept for 4 days in vitro (DIV 4), revealing labeled cells within the cerebellar primordium.

**Figure supplement 2.** FAST DiI was applied to the embryo at embryonic day (E9) and removed after 24 hr, resulting in the visualization of labeled cells within the cerebellar primordium (higher magnification in **Figure 8E, e**).

within the CN is observed by expression of several NTZ markers such as MEIS2 (shown in **Figure 7**) MEIS1, POU3F1, BRN2, and IRX3 as documented in **Kebschull et al., 2024**; **Wu et al., 2022** and *c-Ret* and *Tlx3* shown in **Figure 5—figure supplement 1**, indicating the existence of discrete subpopulations of precursors within the CN. MEIS2 (homeobox transcription regulator) is expressed in the mesencephalic alar plate of both mice (**Cecconi et al., 1997**) and chicks (**Agoston and Schulte, 2009**; **Bobak et al., 2009**). MEIS2 plays a crucial role in the development of cranial nerves and craniofacial

structures (*Machon et al., 2015*). It has been suggested that mesencephalon development is regulated by Meis2 cross-talk with Otx2 (*Agoston and Schulte, 2009*). Notably, the expression of MEIS2 in CN neurons is remarkably distinct, allowing for a clear demarcation of the caudodorsal (SNCA$^-$/MEIS2$^+$) and rostroventral (SNCA$^+$/MEIS2$^+$) regions of the NTZ, whether originating from the RL or not (*Figure 7D–I*). CN neurons derived from the RL require ATOH1 expression, as they are absent in an Atoh1-null mutant (*Wang et al., 2005*; *Machold and Fishell, 2005*; *Zordan et al., 2008*; *Florio et al., 2012*; *Ben-Arie et al., 1997*). In the Atoh1-null cerebellar primordium, the caudodorsal (SNCA$^-$/MEIS2$^+$) subset of CN is absent in the NTZ, while the rostroventral (SNCA$^+$/MEIS2$^+$) population is still present, indicating that the Meis2$^+$ rostroventral subset of the CN in the NTZ develops independently from Atoh1 (*Figure 9*).

What are the potential origins of SNCA$^+$/OTX2$^+$/MEIS2$^+$/p75NTR$^+$/LMX1A$^-$ cells in the NTZ during the early stages of cerebellar development? Our study suggests that the origin of these cells may be guided by a continuous flow of rostroventral cells toward the rostral end (mesencephalon), and caudodorsal cells toward the caudal end (RL) of the cerebellar primordium. It is well established that *Otx2* is required for the development of the fore- and midbrain, while *Gbx2* is necessary for anterior hindbrain development (*Simeone et al., 1992*; *Alvarado-Mallart, 2005*; *Li and Joyner, 2001*; *Joyner et al., 2000*). The expression of OTX2 in the rostral and GBX2 in the caudal neural tube is considered limited to the isthmus (*Alvarado-Mallart, 2005*). However, a study by Martinez et al. on chick/quail chimeras determined that the rostral portion of the cerebellar primordium is located more rostrally in the so-called 'mesencephalic' alar plate (*Martinez and Alvarado-Mallart, 1989*). Recently, *Isl1*-positive cells in the anterior cerebellum were shown to exhibit residual OTX2 protein activity (*Wizeman et al., 2019*). Several other studies have shown that rostral cerebellar development is associated with factors expressed in the caudal mesencephalon, such as engrailed family En1 and -2 (*Joyner et al., 1991*; *Hanks et al., 1995*; *Millen et al., 1995*), and Acp2 (*Bailey et al., 2013*; *Bailey et al., 2014*), which are regulated by the multi-signaling center known as the isthmic organizer (*Martinez et al., 2013*). Furthermore, by employing embryonic cultures in conjunction with DiI labeling, we determined that the source of the rostroventral subset of CN neurons appears to be an external cerebellar germinal zone, possibly originating from the mesencephalon, which is in line with other studies (*Nichols and Bruce, 2006*; *Wizeman et al., 2019*). While our current study hints at the possibility of the caudal mesencephalon serving as a novel extrinsic germinal zone for the cerebellar primordium, further investigation is required. Genetic inducible fate mapping and long-term follow-up studies are essential to gain a comprehensive understanding of the origin, fate, and connectivity of, as well as the role played by the SNCA$^+$/ OTX2$^+$/MEIS2$^+$/p75NTR$^+$/LMX1A$^-$ rostroventral subset of CN neurons in the development of the cerebellum.

In conclusion, our study indicates that the SNCA$^+$/OTX2$^+$/MEIS2$^+$/p75NTR$^+$/LMX1A$^-$ rostroventral subset of CN neurons do not originate from the well-known distinct germinative zones of the cerebellar primordium. Instead, our findings suggest the existence of a previously unidentified extrinsic germinal zone, potentially the mesencephalon.

# Materials and methods
## Animal maintenance

All animal procedures were performed in accordance with institutional regulations and the *Guide to the Care and Use of Experimental Animals* from the Canadian Council for Animal Care (Protocol No.: AC11721 [B2022-001]). For this study, we used embryos from 47 CD1 timed-pregnant mice at E9–18 (total number of embryos was 409). Our approach to sample size was to include enough samples to reach a sufficient level, which was determined empirically (in most of the experiments, sample size was equal to 9 embryos). Timed-pregnant, prostatic acid phosphatase (PAP) mutant (*Pap$^{-/-}$*) (*Zylka et al., 2008*; *Hokin and Hokin, 1959*) mice were used at E12, because they do not express *Snca* (*Pap$^{-/-}$;Snca$^{-/-}$*) and are a valuable experimental tool to assess whether SNCA is required in the development of the CN neurons (*Rahimi-Balaei et al., 2019*). In addition, we studied GFP-tagged Otx2 mouse embryos (from Thomas Lamonerie lab at Université Côte d'Azur), a reporter mouse line in which the Otx2 protein is fused to the fluorescence protein GFP (Otx2$^{Otx2-GFP/+}$; *Fossat et al., 2007*). The animal procedures related to GFP-tagged Otx2 mouse embryos were in accordance with Université Côte d'Azur Institutional Animal Care and Use Committee guidelines. *Atoh1* knockout embryos were

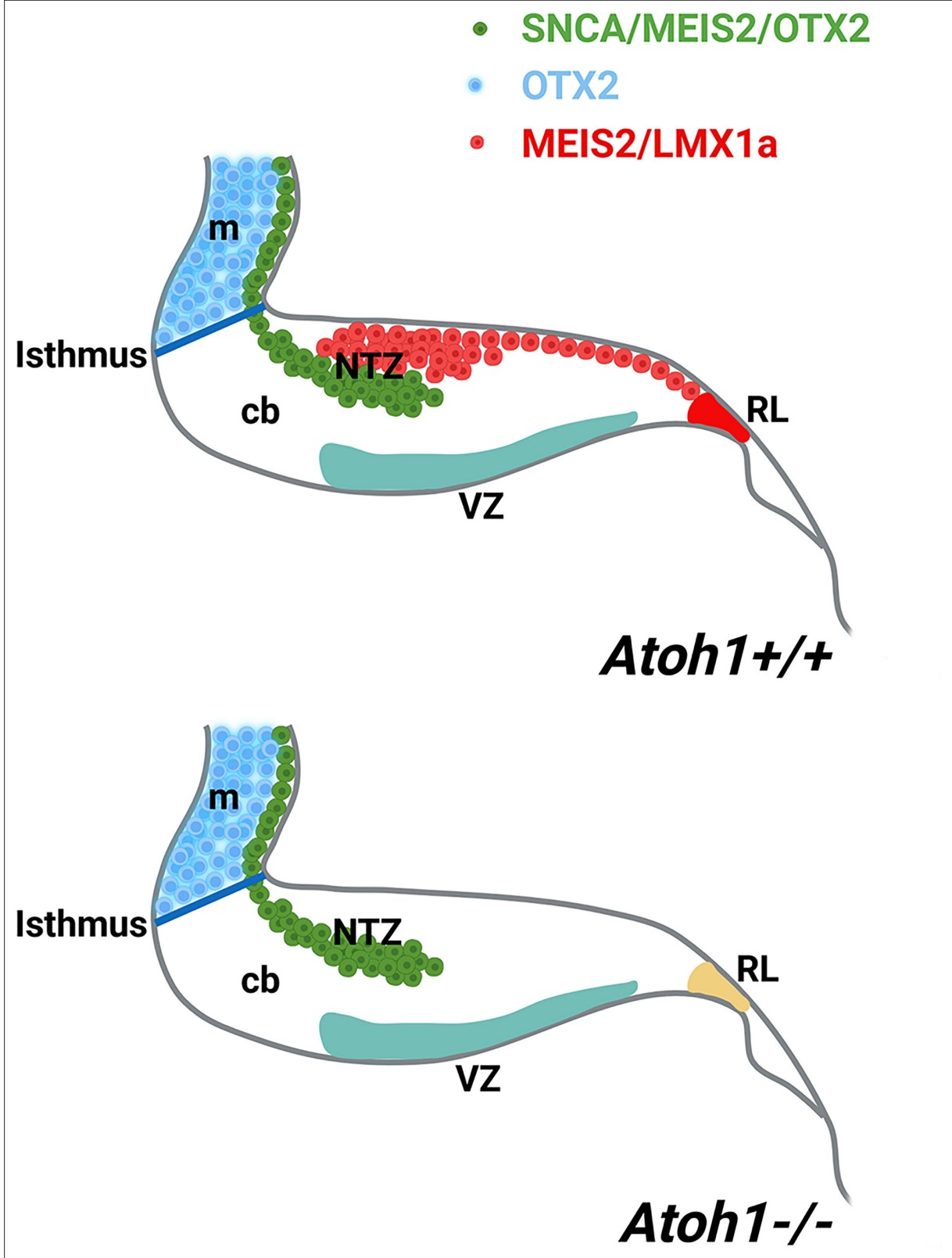

**Figure 9.** A schematic illustration of the developing cerebellum at ~E12 (embryonic day 12), created using BioRender.com. The expression patterns of SNCA, MEIS2, and OTX2 within the cerebellar primordium indicate a subset of cerebellar nuclei (CN) neurons with origins distinct from the rhombic lip (rl). Abbreviations: c, caudal; d, dorsal; m, mesencephalon; ntz, nuclear transitory zone; r, rostral; rl, rhombic lip; v, ventral.

provided by Dr. Huda Zoghbi at the Baylor College of Medicine. The mice were bred, phenotyped, and genotyped (by the Zoghbi lab) according to the previously described protocol (*Wang et al., 2005*). Animal procedures involving Atoh1 knockout mice were in accordance with The Baylor College of Medicine Institutional Animal Care and Use Committee guidelines.

All timed-pregnant CD1 mice were obtained from the Central Animal Care Service, University of Manitoba. Animals were housed at room temperature and relative humidity (18–20°C, 50–60%) on a light:dark cycle (12:12 hr) with free access to food and water. The embryonic age was determined by referencing the morning after the detection of a vaginal plug following overnight mating, considered to be E0.5. CD1 timed-pregnant mice were anesthetized at E9, 10, 11, 12, 13, 14, 15, or 18 (+0.5) (n=47) using 40% isoflurane (USP, Baxter Co. Mississauga, Ontario, Canada), after which embryos were removed and prepared for western blotting, embryonic culture, or FCM or fixed in 4% paraformalde-hyde (PFA) for immunohistochemistry (IHC) or neutral buffered formalin for in situ hybridization (ISH).

## Section IHC

Cryostat sections (20 µm) of 4% PFA fixed samples were utilized for IHC as described in our previous studies (*Bailey et al., 2014*; *Bailey et al., 2013*). Antibody dilutions were used as follows: α-synu-clein 1:500 (sc-69977, Santa Cruz), p75NTR 1:1000 (8238, Cell Signaling), LMX1A 1:500 (AB10533, EMD Millipore Corporation), OTX2 1:1000 (ab114138, Abcam), NAA 1:500 (3A10, Developmental Studies Hybridoma Bank), Meis2 1:500 (ab244267, Abcam), and GFP 1:1000 (1020, Aves Labs). Fluo-rescent detection was performed using antibodies as follows: Streptavidin Alexa Fluor 488 conjugate, Alexa Fluor 568 Goat Anti-Rabbit IgG (H+L), Alexa Fluor 488 Chicken Anti-Mouse IgG (H+L), Alexa Fluor 488 Chicken Anti-Rabbit IgG (H+L), and Alexa Fluor 568 Goat Anti-Mouse IgG (H+L) (S-11223, A-11036, A21200, A21441, A11004, respectively, from Life Technologies), Alexa Fluor 568 Donkey anti-goat (A-11057, Invitrogen), and Alexa Fluor 647 Donkey anti-mouse (A-31571, Invitrogen), all at 1:1000. Detection of peroxidase IHC was performed as described previously (*Rahimi Balaei et al., 2016*; *Bailey et al., 2014*; *Bailey et al., 2013*) using horseradish peroxidase (HRP) conjugated goat anti-rabbit IgG and goat anti-mouse IgG (H+L) antibodies (EMD Millipore Corporation, 12–348 and AP308P, respectively), both at 1:500, and developed with 3,3'-diaminobenzidine solution (DAB, Sigma, St. Louis MO, USA).

## Primary dissociated cerebellar cell culture

Primary cerebellar cultures were prepared from E10 CD1 mice, and maintained for varying DIV (1, 2, 3, 5, and 8), according to published methods (*Shabanipour et al., 2019*). Briefly, the entire cerebellum was removed from each embryo and immediately placed into ice-cold $Ca^{2+}/Mg^{2+}$-free Hank's balanced salt solution (HBSS) containing gentamicin (10 µg/ml) and glucose (6 mM). Next, cerebella were incu-bated at 34°C for 12 min in HBSS containing 0.1% trypsin. After washing, the cerebella were gently triturated with a Pasteur transfer pipette in HBSS containing DNase I (5 U/ml) and 12 mM $MgSO_4$ until the cell mass was no longer visible. Cells were collected by centrifugation (1200 rpm, 4°C for 5 min) and re-suspended in seeding medium (Dulbecco's modified Eagle's medium and F12, 1:1) supple-mented with putrescine (100 µM), sodium selenite (30 nM), L-glutamine (1.4 mM), gentamicin (5 µg/ml), and 10% heat-inactivated fetal bovine serum. Cells were seeded on poly-L-ornithine-coated glass coverslips (12 mm) at a density of $5\times10^6$ cells/ml, and each coverslip was placed into an individual well of a 24-well plate. After 6–8 hr incubation in a $CO_2$ incubator (95% humidity, 37°C, 5% $CO_2$) to accom-plish cell adherence, 500 µl of culture medium supplemented with transferrin (200 µg/ml), insulin (20 µg/ml), progesterone (40 nM), and triiodothyronine (0.5 ng/ml) was added to each culture well. After 7 days, half of the medium in each dish was replaced with fresh medium that was additionally supplemented with cytosine arabinoside (4 µM) and bovine serum albumin (100 µg/ml) (*Bailey et al., 2013*; *Marzban and Hawkes, 2007*; *Marzban et al., 2003*).

## Embryonic cultures and DiI labeling of cells within the mesencephalon

Embryonic cultures were prepared from E9 and E10 CD1 timed-pregnant mice, and maintained for various DIV (4 and 6). Each embryo was removed from the amniotic sac and immediately placed into ice-cold $Ca^{2+}/Mg^{2+}$-free HBSS containing gentamicin (10 µg/ml) and glucose (6 mM). Embryos were placed into 24-well plates in culture medium plus 10% fetal bovine serum and incubated in a $CO_2$ incubator (95% humidity, 37°C, 5% $CO_2$) (*Marzban and Hawkes, 2007*). During incubation, embryos

were monitored every 6 hr to evaluate the heartbeat (as an indicator of survival). On DIV 4 or 6, each well was fixed with 4% PFA and prepared for whole-mount IHC.

For neuronal tracing and labeling, we used FAST DiI crystal (FAST DiI solid; DiIΔ9,12-C18(3), CBS 1,1'-Dilinoleyl-3,3,3',3'-Tetramethylindocarbocyanine, 4-Chlorobenzenesulfonate, D7756, Fisher Scientific). Briefly, FAST DiI was inserted into the mesencephalon at E9 using a sharp-ended needle (30 g). After insertion of FAST DiI, images were captured by a stereomicroscope to assess the location of DiI at DIV 0. After placing the embryos into 24-well plates in culture medium, they were monitored every 6 hr and fixed with 4% PFA on the desired day. Next, whole-mount IHC with NAA (3A10) was performed to visualize neural fiber growth, followed by sectioning and imaging of the DiI-positive cells in the mesencephalon and cerebellar primordium.

### FISH-RNAscope

All of the ISH experiments were carried out using E12 CD1 mice, in which the cerebellar primordium is well established, using RNAscope ACD HybEZ II Hybridization System and RNAscope Multiplex Fluorescent Reagent Kit v2 (Advanced Cell Diagnostics, Hayward, CA, USA). Briefly, embryos were fixed in 10% (vol/vol) neutral buffered formalin at room temperature for 24 hr, dehydrated, and embedded in paraffin. Tissue sections cut at 10 μm thickness were processed for RNA in situ detection according to the manufacturer's user manual. The sequence of the probe used in this study was Mm-*Otx2*-C3 (444381, ACD); and cyanine 3 (NEL744E001KT; TSA Plus, Perkin Elmers, Waltham, MA, USA) was used as the fluorophore.

### Western blot analyses

Equal amounts of total protein were separated by SDS/PAGE in 10–15% precast gels (Bio-Rad, Hercules, CA, USA) and transferred onto PVDF membranes. For western blot analysis, membranes were blocked in 5% nonfat dry milk in TBS containing 0.02% Tween 20 (TBST) for 1 hr at room temperature and subsequently incubated overnight at 4°C with primary antibodies (α-synuclein [sc-69977, Santa Cruz, 1:2000] or p75NTR [8238, Cell Signaling, 1:1000], all dilutions in blocking buffer). After washing with TBST, membranes were exposed to secondary antibodies (HRP conjugated goat anti-mouse IgG [AP308P, Millipore, 1:6000] or HRP conjugated goat anti-rabbit IgG [12-348, Millipore, 1:6000]). Bands were visualized using the enhanced chemiluminescence protocol on scientific imaging film. All bands were normalized to β-actin expression.

### Counting of OTX2-positive cells

To assess the number of OTX2-positive cells, we conducted IHC labeling on slides containing serial sections from E12, E13, E14, and E15 (three embryos were used at each time point). Under the microscope, we systematically counted OTX2-positive cells within the cerebellar primordium. This analysis encompassed a minimum of 10 sections, spread across at least three slides, ensuring comprehensive coverage of OTX2 expression along the mediolateral axis of the cerebellar primordium. For each slide, the counts of OTX2[+] cells from all sections were cumulatively calculated to determine the total number of positive cells per slide. Subsequently, statistical analysis was employed to compare the results obtained at different developmental time points.

### Flow cytometry analysis

After dissociation (see Primary dissociated cerebellar cell culture section), cells were fixed in 4% PFA. Each group consisted of 350,000 cells and all procedures before sorting were performed at room temperature. Primary antibodies used: α-synuclein 1:500 (sc-69977, Santa Cruz), LMX1A 1:1000 (AB10533, EMD Millipore Corporation), and OTX2 1:500 (ab114138, Abcam). Fluorescent detection was performed using antibodies as follows: Alexa Fluor 488 Chicken Anti-Rabbit IgG (H/L) and Alexa Fluor 568 Goat Anti-Mouse IgG (H+L) (A21441 and A11004, respectively, from Life Technologies), both at 1:750. Cells were pelleted by centrifugation, washed, and re-suspended in 200 μl staining buffer (1× PBS contains 1% FBS and 1 mM EDTA). Data were acquired on a CytoFlex-LX flow cytometer (Beckman Coulter) equipped with 355, 375, 405, 488, 561, 638, and 808 Laser lines using CytExpert software, and analyzed with Flow Jo (version 10, Tree Star, San Carlos, CA, USA) at the University of Manitoba flow cytometry core facility. Cellular debris was excluded using forward light scatter/side scatter plot.

## Imaging and figure preparation

For bright-field microscopy, images were captured using a Zeiss Axio Imager M2 microscope and subsequently analyzed with Zeiss Microscope Software (Zen Image Analyses software; Zeiss, Toronto, ON, Canada). For fluorescence microscopy of entire cerebellar sections, a Zeiss Lumar V12 Fluorescence stereomicroscope (Zeiss, Toronto, ON, Canada) equipped with a camera was used; captured images were analyzed using Zen software. For high-magnification fluorescence microscopy, a Zeiss Z1 and Z2 Imager and a Zeiss LSM 700 confocal microscope (Zeiss, Toronto, ON, Canada) equipped with camera and Zen software were used to capture and analyze images. Images were cropped, corrected for brightness and contrast, and assembled into montages using Adobe Photoshop CS5 version 12.

## Statistical analysis

Data on OTX2-positive cell counts are presented as the mean ± standard error of mean (SEM) from n separate experiments. Statistical significance of differences was evaluated by one-way ANOVA followed by post hoc Tukey's multiple comparison test. Differences were considered statistically significant when $p < 0.05$. All statistical analyses were performed using GraphPad Prism 10 software for Windows.

## Acknowledgements

The authors would like to acknowledge Dr. Christine Zhang and Mr. Gerald Stelmack for their technical assistance, and Science Impact (Winnipeg, Canada) for (post-)editing the manuscript. This study was supported by grants from the Natural Sciences and Engineering Research Council (HM: NSERC Discovery Grant # RGPIN-2018-06040), The Children's Hospital Research Institute of Manitoba (HM: CHRIM Grant # 320035), Research Manitoba Tri-Council Bridge Funding Program (HM: Grant # 47955), and University Collaborative Research Program (UCRP).

## Additional information

### Funding

| Funder | Grant reference number | Author |
|---|---|---|
| Natural Sciences and Engineering Research Council of Canada | RGPIN-2018-06040 | Hassan Marzban |
| The Children's Hospital Research Institute of Manitoba | 320035 | Hassan Marzban |
| Research Manitoba Tri-Council Bridge Funding Program | 47955 | Hassan Marzban |
| University Collaborative Research Program (UCRP) | | Hassan Marzban |

The funders had no role in study design, data collection and interpretation, or the decision to submit the work for publication.

### Author contributions

Maryam Rahimi-Balaei, Shayan Amiri, Data curation, Formal analysis, Methodology, Writing - original draft, Writing – review and editing; Thomas Lamonerie, Sih-Rong Wu, Formal analysis, Methodology, Writing – review and editing; Huda Y Zoghbi, Formal analysis, Validation, Writing – review and editing; G Giacomo Consalez, Formal analysis, Writing – review and editing; Daniel Goldowitz, Formal analysis, Validation, Investigation, Writing – review and editing; Hassan Marzban, Conceptualization, Data curation, Formal analysis, Funding acquisition, Investigation, Methodology, Writing - original draft, Project administration, Writing – review and editing

## Author ORCIDs

Maryam Rahimi-Balaei (iD) https://orcid.org/0000-0002-1758-0908
Huda Y Zoghbi (iD) https://orcid.org/0000-0002-0700-3349
G Giacomo Consalez (iD) https://orcid.org/0000-0003-4594-6273
Daniel Goldowitz (iD) https://orcid.org/0000-0003-4756-4017
Hassan Marzban (iD) https://orcid.org/0000-0001-6885-2590

## Ethics

All animal procedures were performed in accordance with institutional regulations and the Guide to the Care and Use of Experimental Animals from the Canadian Council for Animal Care. The animal procedures related to GFP-tagged Otx2 mouse embryos were in accordance with Université Côte d'Azur Institutional Animal Care and Use Committee guidelines. Animal procedures involving Atoh1 knockout mice were in accordance with The Baylor college of Medicine Institutional Animal Care and Use Committee guidelines.Protocol No.: AC11721 (B2022-001).

Reviewer #1 (Public Review): https://doi.org/10.7554/eLife.93778.4.sa1
Reviewer #2 (Public Review): https://doi.org/10.7554/eLife.93778.4.sa2
Author response https://doi.org/10.7554/eLife.93778.4.sa3

# Additional files

## Supplementary files

- MDAR checklist

## Data availability

All data generated or analyzed during this study are included in the manuscript.

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
