## [Editor Report · eLife Assessment]

The authors are interested in the developmental origin of the neurons of the cerebellar nuclei. In this study, they identify a population of neurons with a specific complement of markers that originate in a distinct location from where cerebellar nuclear precursor cells have been thought to originate that show distinct developmental properties. The discovery of a new germinal zone giving rise to a new population of neurons is an exciting finding, and it enriches our understanding of cerebellar development. The **important** claims, better explained in the current version, are well supported by **solid** evidence with the authors using a wide range of technical approaches, including transgenic mice that allow them to disentangle the influence of distinct developmental organizers

---

## [Referee Report · Reviewer #1 (Public Review)]

Summary:

The authors are interested in the developmental origin of the neurons of the cerebellar nuclei. They identify a population of neurons with a specific complement of markers originating in a distinct location from where cerebellar nuclear precursor cells have been thought to originate that show distinct developmental properties. The cerebellar nuclei have been well studied in recent years to understand their development through an evolutionary lens, which supports the importance of this study. The discovery of a new germinal zone giving rise to a new population of CN neurons is an exciting finding, and it enriches our understanding of cerebellar development, which has previously been quite straightforward, where cerebellar inhibitory cells arise from the ventricular zone and the excitatory cells arise from the rhombic lip.

Strengths:

One of the strengths of the manuscript is that the authors use a wide range of technical approaches, including transgenic mice that allow them to disentangle the influence of distinct developmental organizers such at ATOH.

Their finding of a novel germinal zone and a novel population of CN neurons is important for developmental neuroscientists, cerebellar neuroscientists.

Weaknesses:

One important question raised by this work is what do these newly identified cells eventually become in the adult cerebellum. Are they excitatory or inhibitory? Do they correspond to a novel cell type or perhaps one of the cell classes that have been recently identified in the cerebellum (e.g. Fujita et al., eLife, 2020)? Understanding this would significantly bolster the impact of this manuscript.

The major weakness of the manuscript is that it is written for a very specialized reader who has a strong background in cerebellar development, making it hard to read for eLife's general audience. It's challenging to follow the logic of some of the experiments as well as to contextualize these findings in the field of cerebellar development.

---

## [Referee Report · Reviewer #2 (Public Review)]

Summary:

Canonically cerebellar neurons are derived from 2 primary germinal zones within the anterior hindbrain (dorsal rhombomere 1). This manuscript identifies an important, previously underappreciated origin for a subset of early cerebellar nuclei neurons - likely the mesencephalon. This is an exciting finding.

Strengths:

The authors have identified a novel early population of cerebellar neurons with likely novel origin in the midbrain. They have used multiple assays to support their conclusions, including immunohistochemistry and in situ analyses of a number of markers of this population which appear to stream from the midbrain into the dorsal anterior cerebellar anlage.

The inclusion of Otx2-GFP short term lineage analyses and analysis of Atoh1 -/- animals also provide considerable support for the midbrain origin of these neurons as streams of cells seem to emanate from the midbrain. However, without live imaging there remains the possibility that these streams of cells are not actually migrating and rather, gene expression is changing in static cells. Hence the authors have conducted midbrain diI labelling experiments of short term and long term cultured embryos showing di-labelled cells in the developing cerebellum. These studies confirm migration of cells from the midbrain into the early cerebellum.

The authors have appropriately responded to review issues, replacing panels in figures and updating legends and text. They have also appropriately noted the limitations of their work.

---

## [Author Response]

The following is the authors’ response to the previous reviews.

**Reviewer #1 (Recommendations For The Authors):**
(1) I was surprised to see that the Authors have failed to address my major concerns about the paper, which was in the Main text of the Review.Previously I wrote: The major weakness of the manuscript is that it is written for a very specialized reader who has a strong background in cerebellar development, making it hard to read for eLife's general audience. It's challenging to follow the logic of some of the experiments as well as to contextualize these findings in the field of cerebellar development.This has not been addressed. The manuscript has not been substantively changed and it is still written for a very specialized reader rather than a general reader.

We appreciate the respected reviewer’s concern and have made substantial revisions throughout the manuscript to address the points. We have simplified the technical language throughout the manuscript and included additional background information, particularly in the introduction and discussion sections, to better orient general readers. Additionally, we have clarified the logical flow of the experiments by incorporating transitional statements and summaries that explain the purpose and outcomes of each experiment (revisions are highlighted in yellow).

(2) These two have been addressed, although to be honest, I don't think that the cartoon is particularly helpful for a general audience.

Thank you for your feedback. We have replaced the cartoon with a revised version that provides more detailed information to clarify and simplify the origins of cerebellar nuclei from the caudal and rostral ends in both Atoh1+/+ and Atoh1-/- mice. We believe this will make the content more clear and informative for the general audience.

(3) My third recommendation, that they include a section in the Discussion to speculate about what these cells may become in the adult and the existence of multiple cell types with different molecular markers and projection patterns in the nuclei, has also not been addressed.

We apologize for the oversight in the previous revision. We have now added a detailed discussion in the manuscript that speculates on the potential fate of these newly identified cells in the adult cerebellum, suggesting that they may differentiate into excitatory neurons (highlighted on page 9). In addition, as noted in our previous resubmission, further direct evidence is needed from the early population of SNCA+ cells during E9 to E13. This is an ongoing focus of investigation in our lab, where we are currently using SNCA-GFP mice, part of a project for a PhD student in our lab.

**Reviewer #2 (Recommendations For The Authors):**
One small remaining issue: The methods text re cell counts remains confusing: n=3EMBRYOS???"To assess the number of OTX2-positive cells, we conducted immunohistochemistry (IHC) labeling on slides containing serial sections from embryonic days 12, 13, 14, and 15 (n=3 EMBRYOS??? at each timepoint)."

Thank you for this point and we acknowledge that, and we have revised the text in the methods section for clarity. As highlighted on page 11, “The sample size was equal to 9 embryos” and on page 16, “3 embryos were used at each time point”.